# Depth Estimation for Integral Imaging Microscopy Using a 3D–2D CNN with a Weighted Median Filter

**DOI:** 10.3390/s22145288

**Published:** 2022-07-15

**Authors:** Shariar Md Imtiaz, Ki-Chul Kwon, Md. Biddut Hossain, Md. Shahinur Alam, Seok-Hee Jeon, Nam Kim

**Affiliations:** 1School of Information and Communication Engineering, Chungbuk National University, Cheongju-si 28644, Chungcheongbuk-do, Korea; shariar@chungbuk.ac.kr (S.M.I.); kckwon@chungbuk.ac.kr (K.-C.K.); hossain.biddut@chungbuk.ac.kr (M.B.H.); 2VL2 Center, Gallaudet University, 800 Florida Avenue NE, Washington, DC 20002, USA; shahinur.alam@gallaudet.edu; 3Department of Electronics Engineering, Incheon National University, 119 Academy-ro, Yeonsu-gu, Incheon-si 22012, Gyeonggi-do, Korea; icujeon@incheon.ac.kr

**Keywords:** depth estimation, integral imaging microscopy, light-filed microscopy, deep learning, machine intelligence, 3D convolutional neural network

## Abstract

This study proposes a robust depth map framework based on a convolutional neural network (CNN) to calculate disparities using multi-direction epipolar plane images (EPIs). A combination of three-dimensional (3D) and two-dimensional (2D) CNN-based deep learning networks is used to extract the features from each input stream separately. The 3D convolutional blocks are adapted according to the disparity of different directions of epipolar images, and 2D-CNNs are employed to minimize data loss. Finally, the multi-stream networks are merged to restore the depth information. A fully convolutional approach is scalable, which can handle any size of input and is less prone to overfitting. However, there is some noise in the direction of the edge. A weighted median filtering (WMF) is used to acquire the boundary information and improve the accuracy of the results to overcome this issue. Experimental results indicate that the suggested deep learning network architecture outperforms other architectures in terms of depth estimation accuracy.

## 1. Introduction

Integral imaging (II) is a technique that captures and generates entire 3D information in a single capture using 2D arrays of lenses. [1,2,3,4]. The optical microscope has become the most widely used tool in a variety of fields, including biomedical [5], medicine [6], and nanotechnology [7]. Microlens arrays (MLAs) capture and store the depth and the parallax information about microscopic specimens using an integral imaging microscope (IIM) system composed of an objective lens, tube lens, MLA, and a traditional camera [8,9]. The parallax view (i.e., orthographic-view images) is usually reconstructed from the elemental image array (EIA) to provide a 3D model of the specimen [10].

Traditional 2D microscopy primarily improves resolution and generates 2D data that do not recognize the parallax or depth information. It is a serious issue when 3D data are required. Different methods of 3D microscopy have been proposed to overcome this problem, including confocal [11], stereoscopic [12], integral imaging microscopy (IIM), light field microscopy (LFM) [13], and integral imaging holographic microscopy [14]. Depth estimation is a fundamental stage in integral imaging, establishing the way for future LF applications such as 3D reconstruction [15] and augmented reality [16]. A variety of techniques can be used to enhance the depth of field for IIM systems, including amplitude-modulated microlens arrays [17], spatial multiplexing [18,19], bifocal holographic microlens array [20], and a liquid-crystalline polymer microlens array with switchable bifocals [21]. These systems have all shown excellent results. However, the limitations of IIM make it difficult to use various approaches. The image reconstruction is improved by a method that involves an optical device as well as software-based interpolation methods [22]. Despite improving the software-based application of reconstructed images, these conventional methods are not able to produce satisfactory results. Low resolution and inconsistency of brightness of elemental images (EIs) are the main limitations of IIM systems, as well as illumination effects and tiny numerical apertures of the microlens array. According to the theories of geometric and wave-optics, the depth of field (DOF) of the IIM image is even wider than a conventional optical microscope; it is insufficient for a satisfactory viewing experience. In IIM system, the DOF and IIM systems need to be improved. DOF and inconsistency of brightness of EIs have a direct impact on one another. The depth of a specimen can be obtained from the EIA, but extracting the depth map is quite challenging. The depth information may be acquired through active methods (such as depth cameras), passive methods (such as binoculars, muti-view images), or combining the two approaches [23]. Moreover, an optimal solution for depth information accuracy needs to be obtained by applying multiple constraints, such as epipolarity, sequencing, brightness changes, and regularity. Deep learning-based methods have recently been used in IIM systems. In the case that the depth information is accurately estimated from the original 2D image of the specimen, reconstructed image quality might be improved. The purpose of this paper is to propose a method to estimate disparity for IIMs that improves depth accuracy and quality for 3D reconstruction by combining 3D and 2D convolutional neural networks (CNNs) based on multiview EPIs. To improve the robustness of current IIM systems, such as noise and low-quality disparity estimation that result from limited parallax depth, the learning-based network estimates disparity using effective techniques, i.e., image adjustment and depth filtering.

## 2. Background of IIMs and Depth Map

### 2.1. Integral Imaging Microscopy

The IIM system contains a traditional microscope and a microlens array. Light field IIM images are used as input in the proposed work. Figure 1 illustrates the schematic diagram of the IIM system components. A conventional microscopy system lies between the specimen and the intermediate image plane. An expanded visualization of a specimen is captured at the intermediate image plane after a light beam reflected from a specimen passes through the objective and tube lenses. Using the intermediate image plane, MLAs construct the EIA in front of the camera lens (CL). Since the parallax information varies slightly between each elemental image (EA), depth estimation requires geometric analysis to calculate the disparity between the images. EIA is used to reconstruct the orthographic view image (OVI) from the disparity data.

In Equation (1), an object point (*x*, *y*, *z*) is projected onto the EIA plane through the CL and elemental lens (EL). $f_{MLA}$ and $f_{C}$ are the focal lengths of the microlens array and CL, respectively. Furthermore, φ is the pitch of the elemental lens, which denotes the distance between any two elemental lenses. The focal lengths of the objective and tube lenses are $f_{O}$ and $f_{T}$, respectively. There is a distance *g* between the MLA and the camera lens. Lens positions are indicated by *i* and *j*.
(1)$$\begin{array}{l} X_{EI(i,j)}=\frac{f_{MLA}f_{C}(i\times \varphi -x)-f_{C}i\times \varphi (z-f_{MLA})}{\left(g-f_{MLA})(z-f_{MLA}\right)}\\ Y_{EI(i,j)}=\frac{f_{MLA}f_{C}(j\times \varphi -y)-f_{C}j\times \varphi (z-f_{MLA})}{\left(g-f_{MLA})(z-f_{MLA}\right)} \end{array}$$

In this case, the depth between the EL and the camera lens should be calculated using Equation (2):(2)$$\begin{array}{l} \Delta X_{I}=\frac{f_{MLA}f_{C}\varphi (i_{2}-i_{1})}{\left(g-f_{MLA})(z-f_{MLA}\right)}\\ \Delta Y_{I}=\frac{f_{MLA}f_{C}\varphi (j_{2}-j_{1})}{\left(g-f_{MLA})(z-f_{MLA}\right)} \end{array}$$
where *i* and *j* are the lens position, the same as in Equation (1). For each image, this demonstrates the depth information with the viewpoint. The length and depth information are enhanced with an increase in the number of lenses in the MLA. There are a limited number of directional images depending on the EIA. Generating OVI from EIA is a little bit complicated. The first pixel of an EI generates a first OVI, the second pixel of an EI generates a second OVI, and so on. In the same way, the last pixel of the last EI determines the last pixel of the last OVI.

### 2.2. Deep Learning-Based Depth Estimation Method

Deep learning is becoming popular for predicting depth maps from light fields. Generally, deep learning-based depth estimation is performed using epipolar-based approaches by studying the structure of synthetic light-field data [24]. Convolutional Neural Networks (CNNs) have achieved state-of-the-art performances for segmentation [25], super-resolution [26], depth estimation [27,28,29,30], and 3D reconstruction [31] using EPI-based methods. Li et al. [32] introduced an end-to-end fully convolutional network (FCN) for estimating the depth value as nearby pixels in an EPI have a similar linear structure called oriented relation module (ORM). However, these EPI-based approaches only take the EPI features of horizontal and vertical directions, which is insufficient information and decreases the accuracy of the results. Due to the lack of global information, additional optimization processing is also needed. Luo et al. [33] used a CNN architecture to estimate the disparity of each pixel after extracting a set of valid EPI patches from a 4D light field. The output of CNN was processed to post-processing with global constraints. Shin et al. [27] implemented a multi-stream network using fully CNN-based techniques to extract features related to epipolar properties of four viewpoints with horizontal, vertical, and both diagonal directions. For each branch, they combine three convolutional blocks, each of which is usually formed up of ‘Conv-ReLU-Conv-BatchNorm-ReLU’ operations. On the last branch, they apply eight more convolutional blocks of the same type. Heber et al. [28] suggested an end-to-end deep network architecture that included both encoding and decoding modules to predict EPI line-based depth estimation. Shi et al. [34] introduced a learning-based depth estimation system that uses a FlowNet2.0 [35] model to create the initial depth map for stereo pairings. Feng et al. [36] suggested a learning-based two-stream depth estimation method based on a CNN network that learns from nearby pixels. A network based on epipolar segmentation and multi-stream geometry was proposed by Wang et al. [37] for depth estimation. This network uses a sequence layer ‘Conv-Relu-Conv-BN-Relu’. A multi-stream depth network combining cost volume and 3D aggregation was presented by Li et al. [38]. Fsluvegi et al. [39] proposed a fully connected CNN network with two streams as well as augmentation methods for tackling the overfitting issue. However, this is not sufficient to address the occlusion problem. According to Leistner et al. [40], an EPI-Shift network is applied when light field stacks are shifted from wide to narrow baselines, and trained models are used to predict depths from narrow baseline datasets. Even though this method can work for broad-baseline light fields, it is laborious and not suitable for practical applications.

The above learning-based systems still have limitations. In creating the network, those methods mainly evaluate the directed epipolar geometry of light field images, resulting in low depth prediction reliability. In contrast, this paper presents a learning-based network combing the 3D and 2D CNN to predict the depth values from the corresponding horizontal, vertical, and diagonal EPIs. A WMF filter enhances depth prediction by reducing noise and maintaining edge information, which is important for disparity prediction.

## 3. Depth Estimation Methodology

The entire procedure is separated into three parts: image capture through IIM system with camera calibration [41], depth estimation using 3D convolutional neural network, and filtering by weighted median to eliminate depth noise [42]. During the capture process, the specimen is positioned in front of the microscope objective lens. Each microlens is a standalone image source that provides a perspective view image referred to as EI. The EI cannot be applied directly to depth maps. An OVI is constructed using the information provided by the EIA. Hence, the OVI is generated by the EIA, which includes the directional view images. The number of directional images and the resolution are the same as the number of EIs and the resolution, respectively.

A detailed architecture of the proposed IIM depth enhancement system is shown in Figure 2. The EIA is captured by an objective lens, tube lens, and microlens array of an IIM capturing system shown in Figure 1. The ‘gear’ sample is taken as a specimen. This EIA is captured with a camera that is capable of 4028 × 4028-pixel resolution. Using the EIA, the OVI is recreated using a pixel mapping technique [43]. As shown in Figure 2a, the OVI images become steadily dimmer from the center outward due to the properties of the microscopic lighting and MLA. As a result, the images at the boundaries cannot be used to estimate depth, and camera calibration and the region-of-interest (ROI) with a 2964 × 2964-pixel resolution are considered first. The brightness balance between input images is reduced when surrounding OVI are included; however, the parallax view is inadequate to assess depth. In this case, the gap between input directional OVI is taken into consideration. We choose directional images with four intervals from ROI, rather than neighboring images, so that a sufficiently large number of directional-view images can be used. For any object, the brightness distribution of directional images changes progressively from the center to the outer regions of the OVI. In-depth estimation and nonuniform luminance between directional images generates errors and noise. To solve this problem, IIM camera calibration is performed [41]. As a result, the final images are considerably smoother and brighter than the originals, similar to the reference image. As shown in Figure 2b, the directional-view images are then fed into a proposed CNN model that has been pre-trained to evaluate their depth. Finally, postprocessing is necessary for correcting noise during depth estimation. At this stage, in Figure 2c, the WMF approach [44] is used to eliminate internal noise of depth map. A central directional view image is used as a source image in WMF method. In this way, a noiseless depth map can be derived. In Figure 2d, a depth estimate image is acquired when this operation is completed.

### 3.1. Data Generalization

In our proposed model, a representative dataset is required to train the network. A synthetic light-field dataset [24] is applied to the proposed network to reach our goal. Due to the higher angular resolution, light-field images have included a significantly larger quantity of data than images captured with traditional stereo cameras. Several traditional methodologies such as horizontal, diagonal, and multi-direction viewpoint EPIs were used as input data to estimate local disparities. This experiment employs images with four viewpoint angles (0, 90, 45, and 135 degrees) to generate multi-orientation EPIs. The proposed network accepts EPI cuboids with a size of 9 × height × width × 3 as input, suggesting that it is parametric and adjustable to the spatial size as training EPIs are 512 by 512 but test EPIs are 75 by 75. Binary data cuboids [39] are used for training, validation, and testing, where height and width are both 32. During the process of creating the dataset, we use a stride of 13 to iterate through the original image data. Therefore, 38 × 38 = 1444 EPI cuboids were calculated for each scene. As shown in Figure 3, binary file stores each EPI cuboid as a piece of data. Then, 1444 binary files are used for training and testing in each scene.

### 3.2. Network Architecture

As stated previously, our aim is to design a simple and fast convolutional model. The proposed network is divided into four branches as shown in Figure 4. In this design, horizontal and vertical EPI cuboids are used as inputs for the first two branches, and diagonal data cuboids for the last two branches. Each of the preprocesses consists of two steps. First, zero-mean normalization is implemented in batches to the training dataset, revealing the differences across EPI lines and making it easier for the network to learn. Second, when performing 3D convolution, 4 × 4 spatial padding is applied to maintain the spatial dimension of the feature maps. Each branch of the network has four 3D convolutions and a 3 × 3 × 3 kernel with ‘VALID’ padding, stride size ‘1’, and ‘ReLU’ activation. The first convolution generates 32 feature maps, the following two generate 64 feature maps, and the last convolution generates 128 feature maps.

The 2D convolution layer sets acquired from the two streams are combined to create a high-level depiction. The first two and last branches are concatenated and fed into the next 2D convolutional layers to obtain a greater output with minimal loss. The size of the feature maps representing the number of views is decreased to one after the seven convolutions so that a squeezing layer can be applied to remove this dimension and produce 3D data with a size of height × width × 64. As convolutional layers are passed through, the number of feature maps decreases to 128, 64, 32, 32, 16, 8, and 1, respectively, with the kernel size being 3 × 3 and the padding being ‘SAME’. Except for the last convoluted layer, each 2D convoluted layer includes the ‘ReLU’ activation function. A squeezing layer is added at the end of each layer, so the depth map has the dimensions height × width. Finally, two 2D convolution branches are merged into an output layer. A batch size of 32 is used to train the network due to the heavy data and computation involved. For training, set the number of epochs to 2000, the learning rate to 0.0004, and the epochs per decay to 100 with a decay factor of 0.9. After that, the output is passed through the weighted median filtering (WMF) method as shown in Figure 5. The WMF mainly eliminates internal noise while maintaining boundary information, which is essential for depth estimation. A three-channel RGB image should be used as a guidance image for the WMF method for the central directional-view image. A depth estimation image is acquired at the end of this process. A higher-quality image is produced when the WMF method is applied to the initial depth map.

### 3.3. Loss Function

The 4D light-field HCI benchmark [24] dataset contains very precise ground truth values for disparity and performance evaluation. Therefore, a robust loss function should ensure accuracy as well as consistency between the ground truth and the network output. To enforce accuracy for network outputs, most prior studies employed mean absolute error (MAE) between the predicted value (${\hat{y}}_{i}$) and its ground truth ($y_{i}$) as loss function:(3)$$Loss_{MAE}=\frac{1}{N}\sum_{i=1}^{N}\left|\Delta E_{i}\right|$$
where the absolute errors $\Delta E=\left|y_{i}-{\hat{y}}_{i}\right|$ and *N* is the number of errors.

## 4. Experimental Setup and Quality Measurement Metrics

The experimental setup included an infinity-corrected optical system (Olympus BX41TF microscope, Olympus Corporation, Tokyo, Japan) with a 10× magnification. For IIM, a 100 × 100 microlens array is used, with each lens having a diameter of 2.4 mm and a focal length of 125 μm. The photo was taken with a Sony α6000 camera. This sensor has a resolution of 6000 × 4000 pixels and can capture 4028 × 4028 pixels of IIM data.

The model was trained using an Intel Core i7-9800X 3.80 GHz processor and 128 GB memory with an NVIDIA GeForce RTX 2080 Ti GPU running Windows 10 Pro 64-bit. To train and test this network, we utilize the Python programming language in the Pycharm environment using the TensorFlow library. As there are no available datasets for IIM, the network is trained using data from the popular Heidelberg Collaboratory for Image Processing (HCI) 4D light-field benchmark [24], which has 24 scenes, 15 of which were used for training, 4 for validation and the rest for testing. Each scene in the HCI benchmark has 9 × 9 angular view and 512 × 512 spatial resolutions. There are 8 bits of light fields for every scene (9 × 9 × 512 × 512 × 3), camera parameter, and depth information. Disparities can range from −1.5 pixels to 1.5 pixels; however, some scenes have disparities of up to three pixels. Specifically, this benchmark focuses on challenges that present difficulties for depth estimation such as occlusion of boundaries, structures, smooth surfaces, sensor noise, and low textures. Blender was used to render the stratified scenes, and Cycles for the photorealistic scenes. The training loss and validation graph is shown in Figure 6. Tests are performed using the real light-field IIMs image provided by Kwon et al. [42]. The experiment was conducted with three types of real object reconstruction of the OVIs, as shown in Figure 6. Each scene included 53 × 53 directional-view images, with each view containing 76 × 76 pixels.

There are various types of image quality measurement methods to compare the original image and the output image. The root-mean-square error (RMSE) is used to test the HCI dataset. The discrete entropy and power spectrum density (PSD) values are used to analyze our IIM data.

### 4.1. Root-Mean-Square Error

We find the root-mean-square of a set of numbers by squaring them, finding their arithmetic means, and taking their square root.
(4)$$RMSE=\sqrt{\frac{\sum_{i=1}^{n}y_{i}^{2}}{n}}$$
where *y* is the number of observation and *n* is the total number of observations.

### 4.2. Discrete Entropy

In theory and intuitively, it is possible to measure the natural information entropy of the image. As an illustration, one way is to consider an image as a collection of pixels. The discrete entropy (DE) [45] index is used as the quantitative measure of the contrast improvement, which is calculated as follows:(5)$$DE=-\sum_{i=1}^{n}p(x_{i})\times \log_{2}p(x_{i})$$
where *i* is the number of grayscales and *p*(*x_i_*) represents the probability associated with each grayscale. For a single-channel 8-bit image (256 intensity levels), the *p*(*x_i_*) value can be computed as follows:(6)$$p(x_{i})=\frac{x_{i}}{T_{x}}$$

Here, the number of occurrences of intensity level $x_{i}$ and the total number of intensity levels $T_{x}$ are displayed.

### 4.3. Power Spectrum Density

A power spectrum density (PSD) function is a method for assessing image quality without using any reference image [46]. The power spectrum of a signal shows the intensity of a signal at a particular frequency. The PSD can be calculated with the 2D Fourier transform as shown in Equation (7):(7)$$PSD=\log_{10}{\left|F[x(t)]\right|}^{2}$$
where *x*(*t*) is the time-series signal. However, Equation (7) provides continuous spectral data. The PSD value is quantified by calculating the mean value of each spectral power.

## 5. Result and Discussion of the Proposed Depth Enhancement Method

The proposed depth evaluation is carried out using real data from the IIM system. This experiment was conducted with three different types of real objects: a watch gear (Figure 7a), a microchip resistor (Figure 7b), and a flower seedpod (Figure 7c). The three EIA images were created with 76 × 76 EIs, resulting in a resolution of 4028 × 4028 pixels, which is the same as the resolution of the 2D color images as shown in Figure 7 (second column). In Figure 7 (third column), reconstruction of the OVIs included 53 × 53 directional-view images, with each directional-view image being 76 × 76 pixels in size. In this case, the ROIs were taken for each OVI, with 39 × 39 directional images used for depth calculation. As input data, nine by nine images of a star shape are selected at four-image intervals in each of the four directions. Nine images are taken at intervals of four images in each direction to generate the test data.

The HCI dataset is used to estimate disparity maps for synthetic data experiments. To evaluate our method, RMSE values are used. We compared RMSE values with the state-of-the-art algorithms. In these cases, our method is compared with EPI-ORM [32], EPINET [27], MANET [38], and 3D-CNN-LF Depth [39]. The results of the qualitative evaluation (boxes, dino, dots, pyramids, and town) are shown in Table 1.

According to Table 1, the five network models estimate disparity maps based on ground-truth data. In comparison with the actual value, the disparity maps (boxes, dino, and town) of the proposed method are closer to the ground truth. Although the disparity map of the dots and pyramids dataset is quite noisy, it is closer to the ground truth than other datasets.

To evaluate our method for estimating light-field depth, we compared it with state-of-the-art algorithms. In these cases, EPI-ORM [32], EPINET [27], MANET [38], and 3D-CNN-LF Depth [39] were quantified with a discrete entropy (DE) test. All specimens are shown in Figure 7. Since there are no ground-truth values for real objects, DE index and PSD values are used to evaluate the enhanced depth quality. In the depth images (Figure 8), the discrete entropy formula determined that there was better contrast than existing methods.

Figure 9 shows quantitative results, with higher values indicating better results. Compared to existing methods, the proposed CNN-based depth images showed a better contrast, as determined by the discrete entropy formula. The discrete entropy values for the gear, the microchip, and the seedpod were 6.47, 6.41, and 6.82, respectively. In this case, the entropy values for the gear, microchip, and seedpod were (3.76, 3.71, 2.83) for the EPI-ORM, (4.99, 4.76, 5.49) for the EPINET, (5.15, 5.72, 5.85) for the MANET, and (4.66, 5.79, 6.07) for the 3D-CNN-LF Depth method, respectively. Comparisons show that the proposed learning-based disparity calculation method provides a clearer and more accurate 3D depth map than existing methods.

Based on estimated depth map and point cloud interpolation, reconstructed 3D point cloud models were created as shown in Figure 10. Using surface subdivision-based points interpolation, a 3D point cloud of the gear, microchip, and seedpod was created using 2D images and the proposed depth map. In addition to the gear, the microchip, and the seedpod were regenerated using 3D point clouds (562,500 object points for the gear, 577,600 object points for the microchip, and 291,600 object points for the seedpod) and a surface subdivision-based interpolation method [42], which included nearly 10–20-fold more points than the original generation.

Table 2 presents the PSD values for 2D images, central image of directional-view images, and central views for the reconstructed object models in the point cloud of different methods. Therefore, PSD values of 2D corresponding images at high resolution were used to check that both reconstructions of the depth map matched the original measurements. For the gear, microchip, and seedpod, the PSD values of reconstructed 3D images by the proposed depth method were 6.28 dB, 6.03 dB, and 5.95 dB, respectively. In 2D images, the PSD values were 6.47 dB (gear), 6.410 dB (microchip), and 6.420 dB (seedpod). For the central directional view images shown in Figure 6, the PSD values were 5.41 dB, 5.66 dB, and 5.03 dB. These PSD values confirmed that the proposed method provided a clearer 3D presentation based on 3D depth measurement of the three specimens. Furthermore, these results demonstrated that the suggested technique accurately replicated the depth images. Therefore, the proposed CNN beats previous depth estimation methods for the IIM system.

## 6. Conclusions

This paper presents an effective and efficient deep learning-based depth estimation method using a 3D and 2D fully convolutional neural network for IIMs. In the proposed method, geometry structure and edge information are exploited more effectively in the IIM images. This network has the capability of handling photo-realistic depth images and reconstructing 3D images that are virtually identical to the original. The network is much better at resolving noise and occlusion than existing methods. To collect the EIA data, a camera sensor is used in combination with a lens array and a 2D optical microscope. The OVI is then generated from the EIA using the pixel mapping algorithm. Observers achieve a sense of three-dimensionality by viewing directional view pictures of OVI. Multi-stream three-dimensional CNNs are fed directly from the directional view images. As a result, a high-quality depth map image can be obtained.

The proposed method outperforms all current state-of-the-art algorithms. The depth map image generated by this method takes only a few seconds. Furthermore, it indicates that the enhanced depth quality rarely depends on the number of viewing angles. Future research will focus on obtaining more detailed images that are faster to generate and with better depth enhancement. Furthermore, alternative noise reduction approaches will be employed, and existing deep learning algorithms will be compared to our enhanced deep learning model. Another key fact to remember is that the IIM dataset is inadequate. In the future, we will focus on developing algorithms and datasets for the deep learning network.

## Figures and Tables

**Figure 1 sensors-22-05288-f001:**
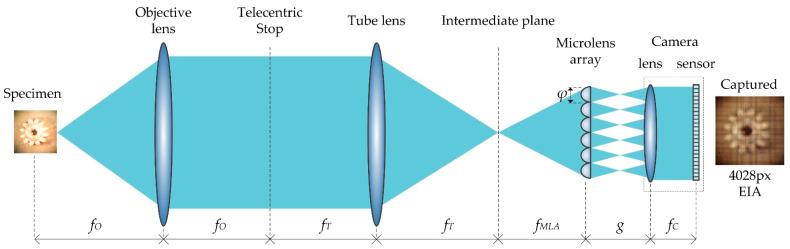
Schematic diagram and image capturing system of an IIM system. Microlens arrays are placed in front of the sensor to capture EIs from an object placed in front of the objective lens and magnified by the tube lens.

**Figure 2 sensors-22-05288-f002:**
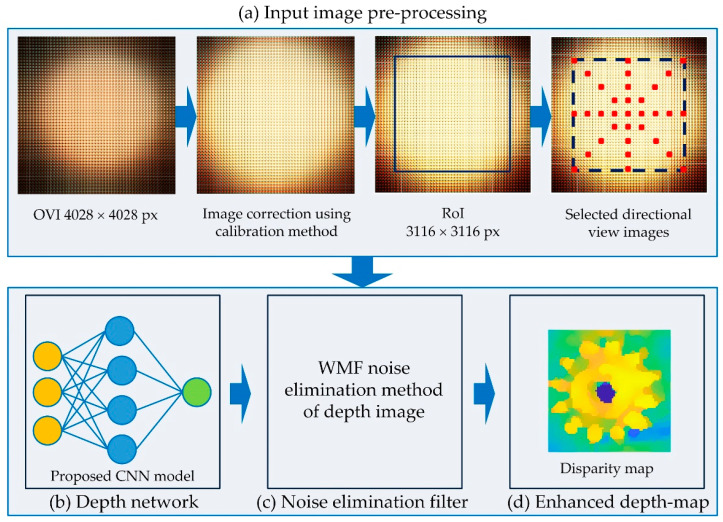
An illustration of the proposed IIM depth enhancement method.

**Figure 3 sensors-22-05288-f003:**
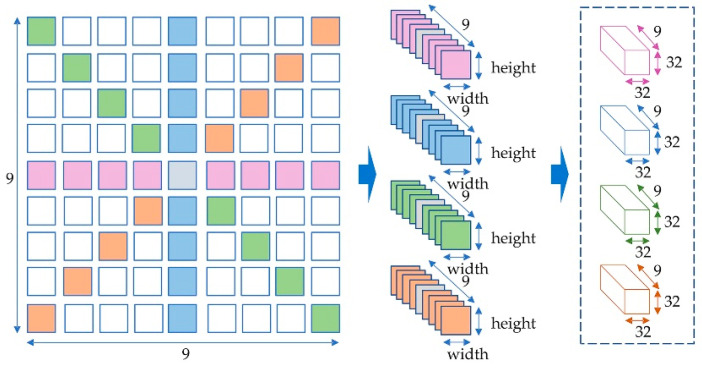
The process of generating EPI cuboids.

**Figure 4 sensors-22-05288-f004:**
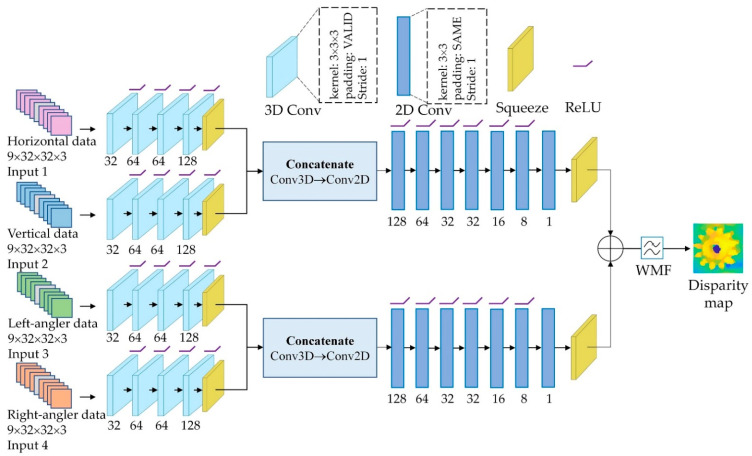
The architecture of the proposed depth estimation network for IIM system.

**Figure 5 sensors-22-05288-f005:**
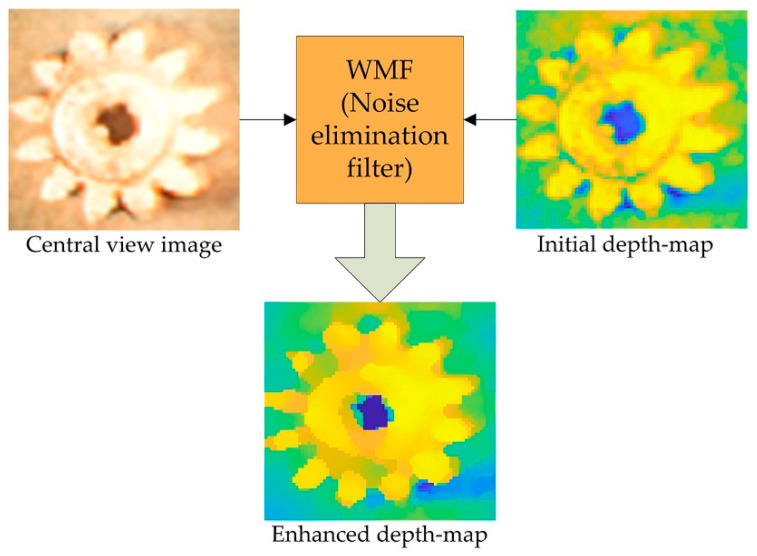
Internal noise reduction using the WMF.

**Figure 6 sensors-22-05288-f006:**
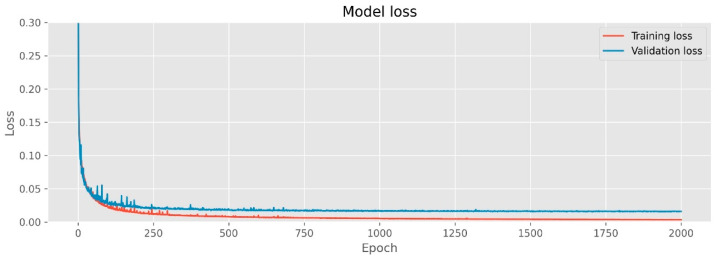
Training loss and validation for the proposed network based on 3D and 2D CNN.

**Figure 7 sensors-22-05288-f007:**
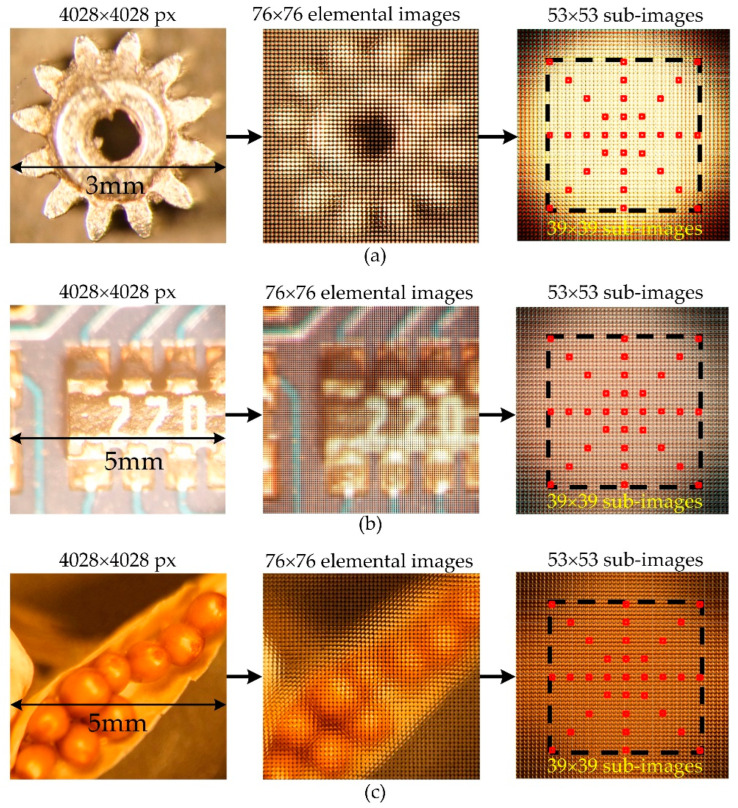
The process of generating test data. Two-dimensional images (left); EIAs (middle); OVIs (right) of the of the (**a**) gear, (**b**) microchip, and (**c**) seedpod.

**Figure 8 sensors-22-05288-f008:**
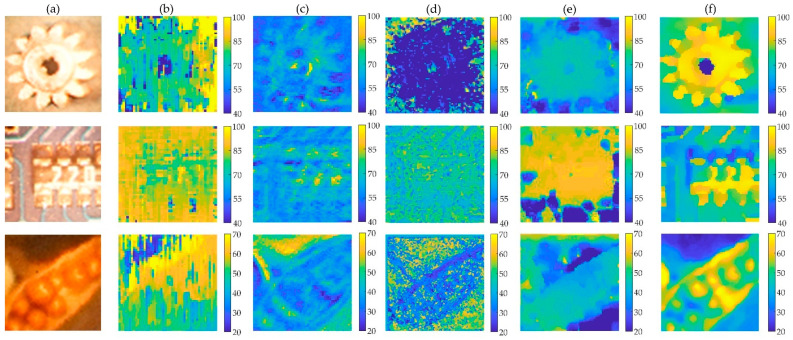
The depth images of three specimen. (**a**) Central directional view image. Depth images generated by (**b**) EPI-ORM [32], (**c**) EPINET [27], (**d**) MANET [38], (**e**) 3D-CNN-LF Depth [39], (**f**) proposed methods, from the same input images.

**Figure 9 sensors-22-05288-f009:**
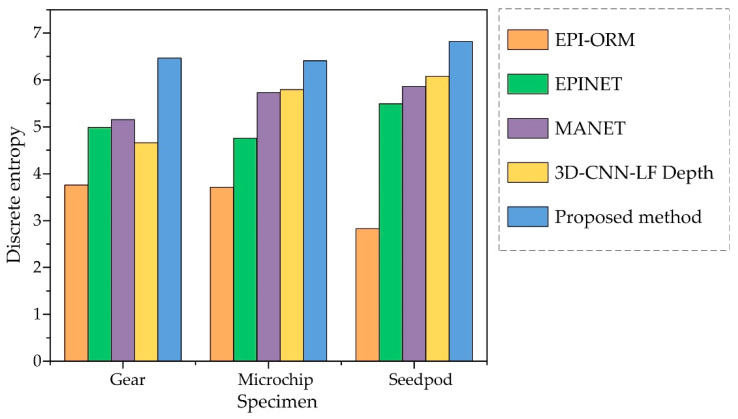
Discrete entropy values for gear, microchip, and seedpod using EPI-ORM [32], EPINET [27], MANET [38], and 3D-CNN-LF Depth [39] proposed depth estimation methods.

**Figure 10 sensors-22-05288-f010:**
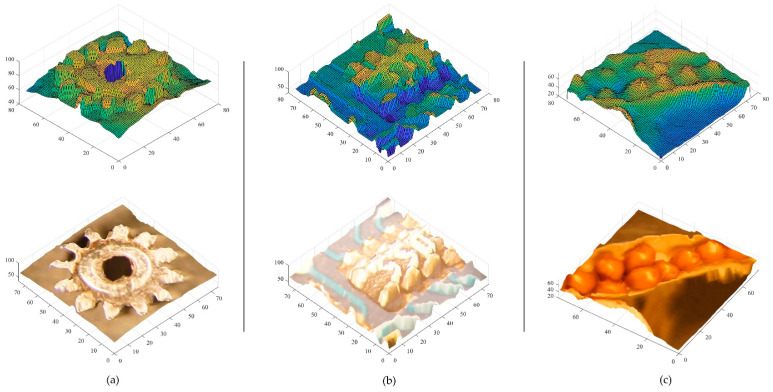
Three-dimensional proposed depth map and reconstructed point cloud models for (**a**) gear, (**b**) microchip, and (**c**) seedpod.

**Table 1 sensors-22-05288-t001:** Comparison RMSE * between the state-of-the-art networks and the proposed model.

Test Images	EPI-ORM [32]	EPINET [27]	MANET [38]	3D-CNN-LF Depth [39]	Proposed
boxes	0.9046	0.7707	1.0247	0.6942	0.6851
dino	0.8601	0.6789	0.9286	0.7035	0.6186
dots	0.8484	0.4624	0.5414	0.6552	0.6526
pyramids	0.7748	0.525	0.9285	0.6487	0.6789
town	0.8301	0.6991	1.2327	0.6608	0.6559

* Lower score represents better performance.

**Table 2 sensors-22-05288-t002:** Quantitative evaluation with the PSD * value for 3D reconstructed image.

Serial	Specimen	2D Image	Center Directional View	Reconstructed Point Cloud				
				Proposed	3D-CNN-LF [39]	MANET [38]	EPINET [27]	EPI-ORM [32]
01	Gear	6.64	5.41	6.28	5.92	5.9	5.89	5.91
02	Microchip	6.81	5.66	6.03	5.67	5.66	5.69	5.69
03	Seedpod	6.88	5.04	5.95	5.42	5.43	5.44	5.47

* For all metrics, higher scores represent better performance.

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
