# Peer review of "Depth Estimation for Integral Imaging Microscopy Using a 3D–2D CNN with a Weighted Median Filter"

_sensors, 2022, doi:10.3390/s22145288_

Round 1

Reviewer 1 Report

The manuscript, entitled "Depth Estimation for Integral Imaging Microscopy using a 3D-2D CNN with a Weighted Median Filter" is well written and presents an interesting approach to a problem of 3D data generation from a 2D image.

1. For consistency, the authors should move the abbreviations 2D and 3D from ln. 28-29 to the abstract
2. There are some errors in the numbering of the equations. The numbering skips from 3 to 7. Also, equation 1 is not placed as a text continuation compared to the others.
3. Some of the pictures look blurred. The authors should include higher resolution images for the final version.
4. For the title of subsection 4.2, the authors should use a full metric name, not an abbreviation
5. This paper should include more detailed information about the dataset and the training/testing procedure, e.g., the division of the dataset into training/validation/testing.
6. Appendix A should be included in the training section and not as a separate section.

Reviewer 2 Report

The paper is clear and the methods are well described.

My main concern is about the training and test processes. The authors state (section 4) that they use the Heidelberg Collaboratory for Image Processing 4D light field benchmark, which has 24 scenes. From them, they use 15 scenes from training, 4 for validation and the rest for testing. The authors must clearly prove that this reduced number of training samples is enough so that the network learns to generalize. Additionally, in section 5, the authors state that they use 3 EIA images to generate the test data. Please clearly clarify which are the tests performed.

Please also clarify how the WMF filter is applied. In fig. 5, the output image substantially differs from the input image, and it does not seem to be the result of a standard WMF filtering to the input image.

Finally, the authors have compared the performance of their algorithm with some other state-of-the-art light field depth estimation algorithms. It would be interesting if the authors include in this comparative evaluation any method that estimates depth from light field microscopy.

Reviewer 3 Report

This study proposes a robust depth map framework based on a combination of 3D and 2D CNN-based deep learning networks to calculate the disparities. I think it is very interesting and should be published after minor revisions for function (9) and figure 8.
